# Vaccines in Gastrointestinal Malignancies: From Prevention to Treatment

**DOI:** 10.3390/vaccines9060647

**Published:** 2021-06-13

**Authors:** Rani Chudasama, Quan Phung, Andrew Hsu, Khaldoun Almhanna

**Affiliations:** 1Division of Hematology-Oncology, Rhode Island Hospital, Alpert Medical School of Brown University, Providence, RI 02903, USA; RChudasama@lifespan.org (R.C.); AHsu2@lifespan.org (A.H.); 2Department of Medicine, Rhode Island Hospital, Alpert Medical School of Brown University, Providence, RI 02903, USA; QPhung@lifespan.org

**Keywords:** gastrointestinal cancer, vaccines

## Abstract

Gastrointestinal (GI) malignancies are some of the most common and devastating malignancies and include colorectal, gastric, esophageal, hepatocellular, and pancreatic carcinomas, among others. Five-year survival rates for many of these malignancies remain low. The majority presents at an advanced stage with limited treatment options and poor overall survival. Treatment is advancing but not at the same speed as other malignancies. Chemotherapy and radiation treatments are still only partially effective in GI malignancies and cause significant side effects. Thus, there is an urgent need for novel strategies in the treatment of GI malignancies. Recently, immunotherapy and checkpoint inhibitors have entered as potential new therapeutic options for patients, and thus, cancer vaccines may play a major role in the future of treatment for these malignancies. Further advances in understanding the interaction between the tumor and immune system have led to the development of novel agents, such as cancer vaccines.

## 1. Introduction

Gastrointestinal (GI) malignancies are some of the most common malignancies and include colorectal, gastric, esophageal, hepatocellular, and pancreatic carcinomas. Overall five-year survival rates for many of these malignancies are low, with many patients presenting with advanced disease. Thus, it is important to continue to investigate and create novel therapeutic interventions to treat these malignancies. Recently, with the advances in immunotherapy in GI malignancies, the development of cancer vaccines has become an important area of interest.

## 2. Role of Vaccines in Cancer

Therapeutic cancer vaccines are a form of immunotherapy that aims to utilize a patient’s own immune system to treat their cancer. In contrast to prophylactic vaccines (such as influenza or pneumococcal vaccines), which are given to healthy patients, therapeutic cancer vaccines are administered to patients who have already been diagnosed with cancer [1]. An effective vaccine would ideally target antigens that are expressed specifically on cancer cells, are necessary for cancer survival, and are highly immunogenic [2]. There are many studies focusing on therapeutic cancer vaccines given their high potential for immunotherapy-based advancement in cancer treatment; however, relatively few cancer vaccines have had clinical success. With further development of this field and the ability to identify additional targets, some researchers have projected that therapeutic cancer vaccines will play a larger role in cancer treatment, serving as a supplement or even alternative to traditional treatment modalities including surgical resection, chemotherapy, and radiation [3].

## 3. Types of Vaccines (Mechanisms)

Although the clinical application of therapeutic cancer vaccines has been relatively limited, it remains an active area of research, with multiple strategies (Figure 1) to deliver these vaccines. Overall cancer vaccines can be organized into: (1) cellular vaccines; (2) protein/peptide vaccines; and (3) genetic (DNA, RNA, and viral) vaccines [3].

Cellular vaccines are comprised of autologous or allogeneic whole tumor cells versus autologous dendritic cells (DCs). These cells are then typically irradiated in order to inactivate further growth and are often combined with an immunostimulatory adjuvant [4,5]. Autologous tumor cell vaccines utilize cells from a patient’s own cancer with the goal of producing an immune response that is unique to that patient’s cancer. An advantage of this type of vaccine is that it could theoretically present all of a patient’s specific tumor-associated antigens (TAAs) to their immune system, which would obviate the need to identify specific TAAs in advance. This approach requires sufficient tumor specimens, which may not necessarily be possible unless a patient’s cancer has progressed to a certain magnitude. In contrast, allogeneic tumor vaccines typically use a few established tumor cell lines, which can help with standardizing and scaling up the production of cellular vaccines [6].

Another type of cellular vaccination involves utilizing immune cells; this field has largely focused on DCs, which are potent antigen-presenting cells (APCs) [7,8]. DC vaccines are created by loading TAAs to a patient’s dendritic cells, which are subsequently treated with immunostimulatory adjuvants. These vaccines are then administered to patients autologously. There has been some clinical success with DC vaccines, particularly in prostate cancer, where it is approved for use in the metastatic, castrate-resistant setting [8].

Protein/peptide vaccines largely target tumor-associated antigens, with the goal of generating an immune response to TAAs that are uniquely or highly expressed in cancer cells compared to normal cells. For example, cancer-testis antigens are a group of TAAs that are encoded by genes that are active in germ cells and reactivated in cancer cells but are normally inactive in adult tissues [9,10]. In contrast, differentiation-associated antigens, such as carcinoembryonic antigen (CEA) and Mucin 1 (MUC-1), are present in both tumor and normal cells but have a higher expression in cancer cells [11,12]. Since TAAs are inherently poorly immunogenic, strategies to enhance protein/peptide vaccines include administration with inflammatory adjuvants and combination with other immune modulators, such as peptide fusion to toll-like receptor agonists [13,14].

An advantage of protein/peptide vaccines is that they can potentially treat any cancer patient with a specific antigen, which would be more resource- and cost-effective than autologous vaccines. However, protein/peptide vaccines require identifying and targeting specific antigens in advance, which can be especially difficult for patients with unique mutations. Protein/peptide cancer vaccines have been studied in diseases like melanoma, prostate cancer, and colorectal carcinoma [15,16]. A gp100 peptide vaccine with interleukin-2 (IL-2) was shown to have a higher response rate than IL-2 alone in a phase III study of patients with melanoma; however, the overall application of protein/peptide vaccines have been limited thus far [17].

Genetic vaccines encompass DNA, RNA, and viral-based vaccines, which aim to introduce genetic material to APCs, resulting in the translation of cancer-specific antigens or antigen fragments [18,19]. An advantage to this type of vaccination is the potential to deliver multiple antigens with a single immunization. DNA vaccines typically utilize bacterial plasmids to deliver DNA inside a patient’s cells. These vaccines have shown improved control of breast carcinoma in murine models; however, these results have not been replicated with humans [18,20]. The overarching challenge to clinical benefit has largely been due to low vaccine uptake efficiency, resulting in low levels of translation. RNA vaccines, which mostly utilize messenger RNA, also attempt to induce the translation of antigens in APCs. An advantage of RNA vaccines compared to DNA vaccines is that their use would theoretically lead to fewer side effects or an autoimmune disease, as they are more rapidly degraded; RNA is also not incorporated into the genome and, therefore, does not introduce further oncogenic potential [21]. As with DNA vaccines, RNA vaccines have also faced issues with effective uptake and subsequent translation of antigens.

Viral-based vaccines have been studied, in part because of the unique potential for viral infections to produce major histocompatibility complex (MHC) class I/II restricted virus-specific peptides in infected cells. Researchers have largely targeted the poxviridae family, adenovirus, and herpes simplex virus (HSV-1) as viral vectors. Poxvirus has been studied extensively due to its ability to incorporate large or multiple transgene inserts. The PROSTVAC phase II clinical study utilized poxviral vaccines targeted at a prostate-specific antigen, which showed improvement in median overall survival (mOS) in men with metastatic prostate cancer [22]. Limitations to the use of viral vaccines include the body’s natural immune response to neutralize viral vectors. As such, some studies have tried to utilize a prime-boost strategy, where a tumor antigen and viral vector are initially administered, followed by a “boost” of the same antigen given by a different viral vector [23].

## 4. Approved Vaccines

While the use of vaccines in cancer has been limited, there have been a few successful instances, as seen in Table 1.

Bacillus Calmette-Guerin (BCG) was first developed as a preventative vaccine from a live-attenuated strain of *Mycobacterium bovis* to prevent *Mycobacterium tuberculosis*. It has also been used in the management of NMIBC since 1977 and received Food and Drug Administration (FDA) approval for use in NMIBC when its intravesicular use was shown to decrease recurrences when compared to surgical resection alone [24]. The Southwest Oncology Group (SWOG) 8507 trial examined the use of intravesicular BCG versus intravesicular doxorubicin in 262 patients with rapidly recurrent stage Ta, T1, or in situ transitional cell carcinoma of the bladder. Treatment with BCG resulted in a higher proportion of complete responses (CR; 70% vs. 34%; *p* < 0.001) and an improved recurrence-free survival (RFS; 22.4 to 10.4 months; *p* = 0.015) [25]. These findings led to the approved use of BCG in the treatment of in situ disease and in preventing relapses in papillary Ta and T1 NMIBC. The role of maintenance therapy was controversial until a follow-up study examining 384 patients who had received intravesicular BCG for induction therapy and were disease-free. The addition of maintenance therapy led to an improvement in median RFS (76.8 vs. 35.7 months; *p* < 0.0001) and is now considered the standard of care [26].

Sipuleucel-T is the first therapeutic cancer vaccine that received FDA approval in 2010 against mCRPC based upon the results of the IMPACT trial. Treatment with this therapeutic cancer vaccine consists of leukapheresis to extract peripheral blood mononuclear cells (PBMCs), particularly dendritic cells, which are then activated ex vivo with the recombinant fusion protein (PA2024). In the phase III D9901 and D9902A trials, a total of 225 patients with mCRPC were randomized in a 2:1 ratio to sipuleucel-T every 2 weeks for 3 total infusions versus placebo. At 36 months, treatment with sipuleucel-T in these two identical trials led to an improved in mOS (23.2 months vs. 18.9 months; hazard ratio (HR) 1.50; 95% confidence interval (CI) 1.10–2.05; *p* = 0.011) [27,28]. The IMPACT trial enrolled 512 patients with mCRPC who were randomized in a 2:1 ratio to biweekly infusions of sipuleucel-T for a total of 3 infusions. The use of sipuleucel-T was associated with improvement in mOS (25.8 vs. 21.7; HR 0.78; 95% CI 0.61–0.98; *p* = 0.03) [8]. These results led to its FDA approval for the use in mCRPC.

Talimogene laherparepvec (T-VEC) is an oncolytic virus derived from the herpes simplex virus type 1 that is used for the treatment of advanced (stage IIIb, IIIc, or IVM1a) melanoma. OPTiM was a phase III trial that enrolled 436 patients with injectable, but unresectable melanoma, who had not received systemic therapy. Patients were randomized in a 2:1 ratio to receive intralesional T-VEC or subcutaneous GM-CSF. The use of T-VEC was associated with a higher durable response rate (DRR; 16.3% vs. 2.1%; *p* < 0.001); overall response rate (ORR; 26.4% vs. 5.7%; *p* < 0.001); and a non-statistically significant improvement in mOS (23.3 vs. 18.9 months; HR 0.79; 95% CI 0.62–1.00; *p* = 0.051) [29]. These results led to T-VEC’s FDA approval for use in advanced melanoma in 2015.

## 5. Vaccines in Gastrointestinal Malignancies

Overall, vaccine-based therapies have had little success in treating GI malignancies. Earlier studies in the 1970s investigated targeting cancer antigens associated with certain mutations and tumor suppressor genes such as RAS and TP53 with the thought that vaccine targets could develop immunogenicity [30].

### 5.1. Colon Cancer

In colorectal cancer (CRC), there have been many different types of cancer vaccines investigated. The use of TAAs has been used in CRC as a potential mechanism for treatment. One of the first TAAs identified was the carcinoembryonic antigen (CEA), which is overexpressed in CRC [31]. In vitro studies showed promising results that a CEA-derived peptide-loaded dendritic cell could induce a CEA-specific cytotoxic T lymphocyte (CTL) response. However, in clinical trials, the efficacy of a CEA peptide vaccine has been low, with a clinical response of less than 17% [32]. The melanoma-associated antigen (MAGE) was first found in melanomas and then was found to be a subgroup of TAAs expressed in a majority of adenocarcinomas. Some studies did find a benefit in MAGE-direct vaccination; however, overall clinical benefit has not been demonstrated [33]. There have been early phase I/II studies that have shown the safety of vaccine targets with some disease response or survival benefit. For example, one phase II study using CRC-specific peptide vaccines in 46 patients with stage III HLAA *2402-positive CRC showed an immune response associated with a survival benefit [34]. A group of peptide vaccines of 13-mer mutant RAS peptide was found to be safe, with immune responses seen in 5 of 11 patients with pancreatic or CRC [35]. However, in a randomized phase III trial of 412 patients with stage II/III colon cancer, there was no significant clinical benefit between the use of an adjuvant autologous tumor cell BCG vaccine with resection versus resection alone [36]. There has been an investigation into combining TAAs as well. For example, a phase I clinical trial combined the ring finger protein 43 (RNF43) with other peptides and showed an 83% disease stability; however, there was no reduction in tumor burden [37]. Personalized peptide vaccines have also been investigated and are achieved by measuring existing peptide-specific CTL in a patient’s blood and, subsequently, vaccinating with CTL-reactive peptides. A phase I trial by Sato et al. showed that about half of the patients had an increase in functional CTL activity after the use of personalized peptide vaccines. However, despite the increase in functional CTL activity, only 20% derived a clinical benefit in the form of partial response or stable disease [38]. The use of neoantigens has also been investigated in CRC, given the high mutational burden in CRC. KRAS mutations play a major role in CRC, and in vaccination trials with peptides made from mutated KRAS, some patients showed a clinical benefit. However, other studies have shown that even with high immune response, this did not correlate with meaningful clinical benefit [39].

### 5.2. Gastric Cancer

In gastric cancer, TAAs have been targeted by vaccines with limited efficacy. In a phase I trial of 14 patients with advanced refractory gastric cancer, there was safety in administration but without clinical efficacy [40]. Another trial of 28 patients with advanced, refractory gastric cancer showed that the combination of dendritic cell vaccine with chemotherapy was safe but did not demonstrate a signal for clinical benefit; furthermore, almost half of the patients experienced disease progression [41]. The gastrin peptide has also been investigated in many trials. In one phase II study, patients with untreated metastatic or unresectable gastric or gastroesophageal adenocarcinoma received G17DT vaccination, which is a 9 amino acid epitope derived from gastrin with chemotherapy. Of the 94 patients, 65 were felt to have an immune response and were found to have a longer time to progression and longer mOS than those who did not have an immune response [42]. One phase I/Ib, open-label, single-arm trial to assess the safety of OTSGC-A24 cancer vaccine in advanced gastric cancer showed in 24 patients that an OTSGC-A24 combined peptide cancer vaccine was well tolerated and had a significant response in cytotoxic T lymphocyte (CTL) induction. However, there was no radiological response seen [43].

### 5.3. Hepatocellular Carcinoma

Cancer vaccines have also not shown much clinical benefit in hepatocellular carcinoma (HCC). Overall, there have not been many successful TAA vaccine trials for hepatocellular carcinoma. The initial rationale for these clinical trials was that the TAAs seen are overexpressed in HCC and are immunogenic in patients. Many clinical trials have also looked at alpha-fetoprotein (AFP) as a potential target, given that it is expressed in up to 80% of HCC patients. AFP-based cancer vaccines were first studies in phase I and II trials by Butterfield et al. [44,45]. These studies showed an anti-AFP-specific T cell response; however, no objective clinical response was seen. Another phase I trial used an AFP peptide vaccine in 5 patients and showed a complete response in 1 patient (CR at 6.7%) [46]. Some phase I trials in peptide vaccines, dendritic cell vaccines, and oncolytic viruses that target certain TAAs have been shown to be tolerated by patients and have some immune response [47]. In a phase I trial by Rizell et al., the use of a dendritic cell vaccine was investigated in combination with sorafenib and showed a significant increase in specific CD8+ T cells in a majority of patients [48]. In one study, 40 patients with advanced HCC were treated with cyclophosphamide followed by GM-GCSF and a telomerase peptide. About half of the patients demonstrated a stable disease 6 months after the initiation of treatment [49]. A recent phase I trial immunized 39 patients with advanced HCC with GPC3 peptide-specific vaccine and did show a GPC3 peptide-specific T cell response in 30 patients and 1 partial response in 1 patient [50]. A phase II study using GPC3 peptide vaccines as adjuvant maintenance therapy in 35 patients with HCC showed efficacy in delaying HCC recurrence at the 1-year, however, not at the 2-year mark [51].

### 5.4. Pancreatic Cancer

There have been various studies in pancreatic cancer looking at the effectiveness of cancer vaccines. A phase II multicenter study that investigated the peptide cocktail vaccine OCV-C01 showed an improvement of median-disease-free survival (DFS) in 30 patients with pancreatic cancer when combined with gemcitabine vs. gemcitabine alone (15.8 vs. 12.0 months) [52]. Thus, this study showed the utility of combined chemotherapy and vaccines to promote increased levels of cancer-specific T cells in immunogenic cancers such as pancreatic cancer. Due to the immunogenic behavior of pancreatic cancer, there have been many studies focusing on antibodies that target tumor cells. For example, vaccines that target TAAs have been used for therapeutic interventions, specifically towards CA 19-9, which is a TAA that is highly expressed on pancreatic cancer cells. Weitzenfeld et al. studied CA 19-9-targeted antibodies and produced them from the serum of CA 19-9/keyhole limpet hemocyanin (KLH)-vaccine-immunized patients. They then used these antibodies in mice to successfully protect mice from pancreatic cancer progression [53]. Some have hypothesized that CA 19-9-targeted vaccines could be used in clinical practice [54]. Besides TAAs, tumor-specific antigens (TSAs) can also be used as antigens expressed on pancreatic cancer cells and can have increased mutations due to their genetic instability. A phase 1b trial was conducted to determine the feasibility of novel autologous DCs pulsed with personalized TSA peptides in pancreatic cancer patients and showed an increased response in CD4^+^ T cells [55]. This study showed that using TSA peptides in vaccines can generate a highly specific immune activation and response against pancreatic cancer cells. Another category of cancer vaccines is the use of DC. There has been data suggesting that using DC vaccines in pancreatic cancer could help inhibit pancreatic cancer metastases through the use of intraperitoneal injection of DC vaccines [56]. Rong et al. conducted a phase I pilot trial which investigated the MUC1-peptide DC vaccine in metastatic pancreatic cancer patients. They found that the vaccine increased the immunologic response to the tumor antigen MUC1 without significant toxicity [57]. Another study showed the benefit of the combination of the peptide-pulsed DC vaccine with the toll-like receptor (TLR)-3 agonist poly-ICLC as a route to inhibit metastases from pancreatic cancer by increasing CD8^+^ T cells [58].

## 6. Vaccines in Prevention (HBV and HPV)

Vaccinations against cancer-causing infections are important to cancer prevention for specific malignancies. Data estimate that 10% of yearly cancers worldwide are caused by viral infections [59]. Vaccinations against hepatitis B virus (HBV) and certain genotypes of human papillomavirus (HPV) currently exist and show efficacy in the prevention of cancers.

HBV vaccines have been available for many years, and certain countries with high rates of endemic HBV infection have been shown to have associated high rates of HCC. In these countries, implementing universal infant HBV vaccination policies has helped reduce HBV infection, which led to a reduction in HCC incidence and mortality [60]. The impact of preventive vaccination programs in combating cancer was exemplified in the HBV immunization program started in Taiwan in 1982. This program initially targeted infants of HBV-infected mothers, then broadened to all infants, and eventually developed into a universal vaccination program leading to significant reductions in the incidence of HCC [61].

## 7. Challenges and Reasons for Inefficacy

In general, there are many challenges facing T-cell-based cancer immunotherapy and cancer vaccines, specifically in GI malignancies. Low immunogenicity is related to aging, as many GI malignancies are diagnosed in older patients or immune cell exhaustion after many lines of previous anti-cancer treatment [62]. The high disease burden and the immunosuppressive tumor microenvironment are additional challenges to these therapeutic agents. Cancer cells can progress through evolving immunosuppressive mechanisms that allow the cells to escape immune attacks [63]. In many gastrointestinal malignancies, such as HCC and gastric cancer, increased inflammation is driven by immune cell infiltration and determines antigenicity [64].

It is also difficult to develop cancer vaccines against TAAs, as these are self-proteins abnormally expressed by cancer cells. B and T cells that recognize these antigens have been removed by the immune system as a part of the central and peripheral tolerance. Thus, targeting TAAs may damage the normal cell expression. This is seen in chimeric antigen receptor-engineered (CAR) T cell therapy targeting CEA, leading to severe colitis in colon cancer patients because the antigen is normally expressed in the intestinal tissue [2,65].

Neoantigens are immunogenic- and cancer-specific; thus, the majority are unique to an individual patient’s cancer. This requires personalized therapy, which can be technically complex and requires significant resources and time, thus further limiting its availability to patients. Many patients with gastrointestinal malignancies present at an advanced stage and thus need timely treatment. Personalized GI cancer vaccines many times take too long to develop to be able to adequately treat certain patients [66]. Another potential challenge, specifically seen in colorectal cancers that are MSI high, is the ability to obtain the whole mutational profile of the tumor in order to develop personalized GI cancer vaccines. Tumor sequencing can only reveal mutations of a subset of cancer cells and only at a specific time point. Thus residual metastatic cells may differ in their mutational status and can lead to the inefficacy of GI cancer vaccines [67].

Cancer vaccines specifically may have been inefficient thus far due to many factors related to the immune system. Patients with GI cancers are felt to not be immunologically naïve and have antigen-specific tolerance to cancer. Immune tolerance involves various T lymphocytes that are either immunogenic or tolerogenic. Therapeutic vaccines increase both types of T lymphocytes, and thus, might amplify cells that are involved in both tumor tolerance and rejection, which nullifies the therapeutic efficacy. In cancer patients, the immune system is not naïve and thus is ready for tumor-tissue-protecting responses. In the prophylactic setting, such as the HBV vaccine, these immune responses help confer protection; however, in the therapeutic setting, the vaccine-induced immune response fails to be clinically beneficial due to the tumor microenvironment comprised of tolerogenic lymphocytes that have either infiltrate into or are in the vicinity of the tumor. Patients with cancer differ from patients without cancer in not only the immune environment but also the presence of a tumor that can act as a de novo organ and maintain its own tissue homeostasis [68,69].

Future studies may be able to utilize cancer vaccines to initiate a local immune response in the tumor that then allows an even more effective anti-cancer immunotherapy response. Table 2 highlights the current clinical trials that are actively recruiting patients to further investigate vaccines that target GI malignancies.

## Figures and Tables

**Figure 1 vaccines-09-00647-f001:**
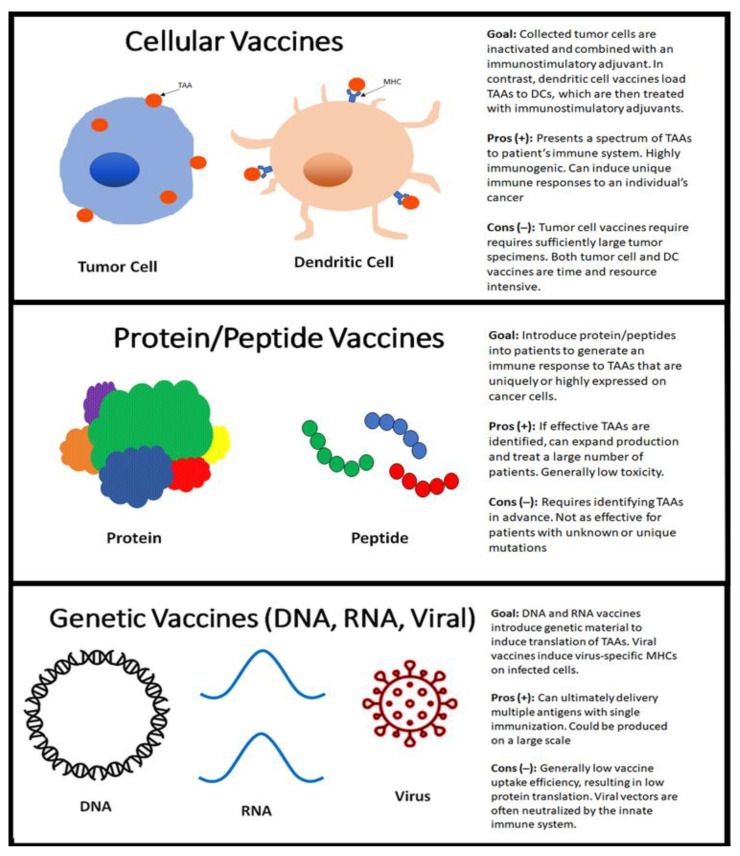
Mechanisms of vaccines: TAA, tumor associated antigens; DC, dendritic cell; DNA, deoxyribonucleic acid; RNA, ribonucleic acid; MHC, major histocompatibility complex.

**Table 1 vaccines-09-00647-t001:** Approved Vaccine-Based Therapies in Malignancies.

Author (Date)—Study Name	Treatment Regimen	Total Patients	Patient Population	ORR/CR	mRFS (Months) HR, *p*-Value	mOS (Months) HR, *p*-Value
Lamm et al., (1991), SWOG 8507 [25]	Intravesicular BCG vs. intravesicular doxorubicin	262	Ta, T1, in situ transitional cell carcinoma of bladder	NR/70%	22.4 vs. 10.4	NR
				NR/34%	NR, *p* = 0.015	
Lamm et al., (2000), SWOG 8507 [26]	Maintenance intravesicular BCG vs. placebo	384	Ta, T1, in situ transitional cell carcinoma of bladder	NR	76.8 vs. 35.7	NR
					NR, *p* < 0.0001	
Small et al., (2006), IMPACT [28]	Sipuleucel-T vs. placebo	512	Metastatic hormone-refractory prostate cancer	NR	11.7 vs. 10.0	25.9 vs. 21.4
					NR, *p* = 0.052	NR, *p* = 0.01
Andtbacka et al., (2015) [29]	Talimogene laherparepvec vs. GM-CSF	436	Unresectable stage IIIB/C and stage IV melanoma	26%/10.8%	NR	23.3 vs. 18.9
				5.7%/<1%		HR 0.79, *p* = 0.51

NR: not reported; NE: not estimable; GM-CSF: granulocyte macrophage colony-stimulating factor.

**Table 2 vaccines-09-00647-t002:** Current recruiting clinical trials for cancer vaccines in GI malignancies.

GI Malignancy	Trial Name	Intervention/Study Arms
Gastric Cancer	Safety and Efficacy Study for MG-7-DC Vaccine in Gastric Cancer Treatment [70]	DC vaccine DC vaccine + CTL DC vaccine + Sintilimab Injection (PD-1 monoclonal ab)
	Nivolumab, Ipilimumab, and OTSGC-A24 Therapeutic Peptide Vaccine in Gastric Cancer—a Combination Immunotherapy Phase Ib Study [71]	OTSGC-A24 + nivolumab OTSGC-A24 + nivolumab + ipilimumab
	A Study of IMU-131(HER-Vaxx) and Chemotherapy Compared to Chemotherapy Only in Patients With HER2 Positive Advanced Gastric Cancer [72]	IMU-131 + Cisplatin + 5-FU or Capecitabine IMU-131 + Cisplatin + 5-FU or Capecitabine or Oxaliplatin and Capecitabine Cisplatin + 5-FU or Capecitabine or Oxaliplatin and Capecitabine
Colorectal Cancer	DC Vaccine in Colorectal Cancer [73]	DC vaccine
	Intratumoral Influenza Vaccine for Early Colorectal Cancer [74]	Influenza Vaccine
	Vaccination With Autologous Dendritic Cells Loaded With Autologous Tumour Homogenate After Curative Resection for Stage IV Colorectal Cancer [75]	Autologous dendritic cells loaded with autologous tumour homogenate + Interleukin-2 (IL2)
	GVAX for Colorectal Cancer [76]	GVAX
	A Trial of Perioperative CV301 Vaccination in Combination With Nivolumab and Systemic Chemotherapy for Metastatic CRC [77]	mFOLFOX6 + nivolumab nivolumab + MVA-BN-CV301 + FPV-CV301
	Phase 1b Study to Evaluate ATP128, With or Without BI 754091, in Patients With Stage IV Colorectal Cancer [78]	ATP128 + BI 754091
	Trial of PalloV-CC in Colon Cancer [79]	PalloV-CC
HCC	DNAJB1-PRKACA Fusion Kinase Peptide Vaccine Combined With Nivolumab and Ipilimumab for Patients With Fibrolamellar Hepatocellular Carcinoma [80]	DNAJB1-PRKACA peptide vaccine + Nivolumab + Ipilimumab
	“Cocktail” Therapy for Hepatitis B Related Hepatocellular Carcinoma [81]	MSDCV with radical surgery therapy Radical surgery therapy MSDCV with TACE therapy TACE Therapy MSDCV with Sorafenib or Lenvatinib Sorafenib or Lenvatinib
	GNOS-PV02 Personalized Neoantigen Vaccine, INO-9012, and Pembrolizumab in Subjects With Advanced HCC [82]	GNOS-PV02 + INO-9012 + Pembrolizumab
Pancreatic Cancer	Neoantigen Peptide Vaccine Strategy in Pancreatic Cancer Patients Following Surgical Resection and Adjuvant Chemotherapy ([83])	Neoantigen Peptide Vaccine + poly IC:LC
	Clinical Trial on Personalized Neoantigen Vaccine For Pancreatic Cancer ([84])	Personalized neoantigen vaccine
	A Trial of Boost Vaccionations of Pancreatic Tumor Cell Vaccine ([85])	Neo vaccine Neo vaccine + single dose cyclophosphamide Neo vaccine + metronomic cyclophosphamide
	Neoadjuvant/Adjuvant GVAX Pancreas Vaccine (With CY) With or Without Nivolumab and Urelumab Trial for Surgically Resectable Pancreatic Cancer ([86])	Cyclophosphamide + GVAX Cyclophosphamide + GVAX + nivolumab Cyclophosphamide + GVAX + nivoulumab + Urelumab
	GVAX Pancreas Vaccine (With CY) in Combination With Nivolumab and SBRT for Patients With Borderline Resectable Pancreatic Cancer ([87])	Cyclophosphamide + nivolumab + GVAX + SBRT
	Study of Personalized Tumor Vaccines (PCVs) and a PD-L1 Blocker in Patients With Pancreatic Cancer That Can be Treated With Surgery ([88])	Atezolizumab + RO7198457 + mFOLFIRINOX
	Study of CRS-207, Nivolumab, and Ipilimumab With or Without GVAX Pancreas Vaccine (With Cy) in Patients With Pancreatic Cancer ([89])	Cyclophosphamide + Nivolumab + Ipilimumab + GVAX + CRS-207 Nivolumab + Ipilimumab + CRS-207
	DC Vaccine in Pancreatic Cancer ([90])	DC vaccine
	Epacadostat, Pembrolizumab, and CRS-207, With or Without CY/GVAX Pancreas in Patients With Metastatic Pancreas Cancer ([91])	Epacadostat + Pembrolizumab + CRS-207 + Cyclophosphamide + GVAX Epacadostat + Pembrolizumab + CRS-207
	Personalized Vaccine With SOC Chemo Followed by Nivo in Pancreatic Cancer ([92])	PEP-DC vaccine + nivolumab + gemcitabine/capecitabine
	Trial of Neoadjuvant and Adjuvant Nivolumab and BMS-813160 With or Without GVAX for Locally Advanced Pancreatic Ductal Adenocarcinomas.([93])	SBRT + Nivolumab + CCR2/CCR5 dual antagonist + GVAX SBRT + Nivolumab + CCR2/CCR5 dual antagonist
	Maintenance Therapy With OSE2101 Vaccine Alone or in Combination With Nivolumab, or FOLFIRI After Induction Therapy With FOLFIRINOX in Patients With Locally Advanced or Metastatic Pancreatic Ductal Adenocarcinoma (TEDOPAM) ([94])	FOLFIRI OSE2101 vaccine OSE2101 vaccine + Nivolumab

DC, dendritic cell; CTL, cytotoxic lymphocyte; 5-FU, fluorouracil; mFOLFOX, modified leucovorin, fluorouracil, oxaliplatin; MVA-BN, modified vaccinia Ankara-Bavarian Nordic; BI, Boehringer Ingelheim; PalloV-CC, Particle-delivered, Allogeneic Tumor Cell Lysate Vaccine for Colon Cancer; MSDCV, Multiple Signals loaded Dendritic Cells Vaccine; TACE, Trans-arterial Chemoembolization; INO, Inovio; Poly LC:IC, carboxymethylcellulose, polyinosinc–polycytidlyic acid, and poly-L-lysin double-stranded RNA; SBRT, Stereotactic Body Radiation; mFOLFIRINOX, modified leucovorin, fluoruracil, irinotecan, oxaliplatin; PEP, personalized peptides; FOLFIRI, folinic acid, irinotecan, 5-FU.

## Data Availability

Not applicable.

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
