# Peer review of "Vaccines in Gastrointestinal Malignancies: From Prevention to Treatment"

_vaccines, 2021, doi:10.3390/vaccines9060647_

Round 1

Reviewer 1 Report

The manuscript "Vaccines in Gastrointestinal Malignancies: From Prevention to Treatment" was written to review the progress of the vaccines in Gastrointestinal malignancies. The manuscript introduced several cancer vaccines and the mechanisms of different types of cancer therapeutic vaccines.  The manuscript did not keep the focus on the vaccines in gastrointestinal malignancies. There is only one part of the paper (Vaccines in Colon, Gastric and Hepatocellular Carcinoma) that is related to the vaccines in gastrointestinal malignancies. The content of this part did not reflect the aim of "the prevention and treatment" The content included in the part of "Vaccines in Colon, Gastric and Hepatocellular Carcinoma" is not enough to be a review paper yet. I would suggest 

  1. Keep the focus on vaccines in gastrointestinal malignancies
  2. include more content of the current studies of vaccines in gastrointestinal malignancies in the paper to provide a valuable review of this field. There are over 6000 publications in PMC when searching "vaccines in gastrointestinal malignancies", It should have enough content to review. 

Author Response

  • Comments from Reviewer 1:
    1. Keep the focus on vaccines in gastrointestinal malignancies
      • We have added additional studies/data on GI malignancy vaccines and decreased discussion on HPV/cervical cancer.
    2. include more content of the current studies of vaccines in gastrointestinal malignancies in the paper to provide a valuable review of this field. There are over 6000 publications in PMC when searching "vaccines in gastrointestinal malignancies", It should have enough content to review.
      • We have included additional studies in gastric, HCC and an additional section on pancreatic cancers. We have added a section on clinical trials in pancreatic cancer.

Reviewer 2 Report

The paper comprehensively introduces a set of information of cancer vaccines. They summarize the recent progress and the therapeutic efficacy in cancer vaccines in not only gastrointestinal malignancies but also other types of cancers. However, the contents are not so impressive and the manuscript is poorly prepared.   

Specific comments:

1) Contrary to the title ‘Vaccines in Gastrointestinal Malignancies, the description on other types of cancers, melanoma, prostate cancer and cervical cancer, is too much.

2) They listed BCG therapy and oncolytic virus therapy Talimogene laherparepvec (T-VEC) in Tables 1. Are these therapies cancer vaccines? They are supposed as biotherapies but not cancer vaccines.

3) There is no explanation on Table 2. Where are the references of 69 – 81? How about the references 66 – 68?

4) Vaccines against pancreatic cancer are not listed in Tables 2.

Author Response

  • Comments from Reviewer 2:
    1. Contrary to the title ‘Vaccines in Gastrointestinal Malignancies, the description on other types of cancers, melanoma, prostate cancer and cervical cancer, is too much.
      • We have decreased discussion on HPV in cervical cancer.
    2. They listed BCG therapy and oncolytic virus therapy Talimogene laherparepvec (T-VEC) in Tables 1. Are these therapies cancer vaccines? They are supposed as biotherapies but not cancer vaccines.
      • Title adjusted to indicate - Approved vaccine based therapies
    3. There is no explanation on Table 2. Where are the references of 69 – 81? How about the references 66 – 68?
      • Explanation on table 2 included in last section of paper. We updated our references.
    4. Vaccines against pancreatic cancer are not listed in Tables 2.
      • These have now been included.

Round 2

Reviewer 1 Report

The revised version is better. but again, I would like to suggest staying focus on only Gastrointestinal Malignancies. as indicated in the title.

  1. Type of the vaccine part. It is OK to introduce the type of vaccine, but this is not the major focus. so this part can be shorter or merge into the part to introduce each GM vaccine.
  2. Approved vaccines. I do not think this part is necessary for this review. or it can be briefly mentioned in the introduction or just mentioned the vaccine for Gastrointestinal Malignancies. So table 1 is not necessary as well.
  3.  Vaccines in Gastrointestinal Malignancies. This part is good now.
  4. Vaccines in prevention. HBV is related, but HPV is not related to Gastrointestinal Malignancies. so the HPV part can be deleted.
  5. Challenges and reasons for inefficacy. I would not just talk in general reason, but talk about the challenges of each type/vaccine that treat or prevent Gastrointestinal Malignancies mentioned in the part 3. you may reference the reason of vaccine efficacy of other cancer here to suggest the possible reason of GM vaccine efficacy.

Author Response

  1. Type of the vaccine part. It is OK to introduce the type of vaccine, but this is not the major focus. so this part can be shorter or merge into the part to introduce each GM vaccine.
    • Have taken out some parts to shorten this section
  2. Approved vaccines. I do not think this part is necessary for this review. or it can be briefly mentioned in the introduction or just mentioned the vaccine for Gastrointestinal Malignancies. So table 1 is not necessary as well.
    • Have decreased some parts of discussion, we still feel the table is useful to outline approved therapies as background information
  3. Vaccines in Gastrointestinal Malignancies. This part is good now.
  4. Vaccines in prevention. HBV is related, but HPV is not related to Gastrointestinal Malignancies. so the HPV part can be deleted.
    • HPV discussion has been taken out
  5. Challenges and reasons for inefficacy. I would not just talk in general reason, but talk about the challenges of each type/vaccine that treat or prevent Gastrointestinal Malignancies mentioned in the part 3. you may reference the reason of vaccine efficacy of other cancer here to suggest the possible reason of GM vaccine efficacy.
    • Have focused more on GI malignancies

Reviewer 2 Report

The authors revised the manuscript to my comments. The revised manuscript may provide information on antic-cancer vaccine for gastrointestinal malignancies.

Author Response

Thank you. We appreciate your time. 

Round 3

Reviewer 1 Report

As this is a review paper, it should include the most updated progress on the topic and learn from the deepened discussion from experts.

The 1st revised version added more content of the vaccines of Gastrointestinal Malignancies. which is a great improvement.

The 2nd revised version did not improve much. For example

  1. line 290, add "specifically in GI malignancies" does not mean the sentence is really discussing the challenges specific to the GI  cancer vaccine
  2. same comments for line 316.

I would suggest the author dig into the more current studies and clinical trials to broaden the content and deepen the discussion.

Author Response

Have added additional discussion regarding challenges for GI vaccines